# Class III Peroxidases in Response to Multiple Abiotic Stresses in *Arabidopsis thaliana* Pyrenean Populations

**DOI:** 10.3390/ijms23073960

**Published:** 2022-04-02

**Authors:** Ali Eljebbawi, Bruno Savelli, Cyril Libourel, José Manuel Estevez, Christophe Dunand

**Affiliations:** 1Laboratoire de Recherche en Sciences Végétales, Université de Toulouse, CNRS, UPS, INP, 31326 Toulouse, France; ali.eljebbawi@univ-tlse3.fr (A.E.); bruno.savelli@univ-tlse3.fr (B.S.); cyril.libourel@univ-tlse3.fr (C.L.); 2Fundación Instituto Leloir and IIBBA-CONICET, Av. Patricias Argentinas 435, Buenos Aires C1405BWE, Argentina; jestevez@leloir.org.ar; 3Centro de Biotecnología Vegetal, Facultad de Ciencias de la Vida, Universidad Andres Bello, Santiago CP 8370146, Chile; 4ANID—Millennium Science Initiative Program—Millennium Institute for Integrative Biology (iBio) Millennium Nucleus for the Development of Super Adaptable Plants (MN-SAP), Santiago CP 8370146, Chile

**Keywords:** class III peroxidases, abiotic stress, natural populations, root development, RNA-seq, ROS

## Abstract

*Class III peroxidases* constitute a plant-specific multigene family, where 73 genes have been identified in *Arabidopsis thaliana*. These genes are members of the reactive oxygen species (ROS) regulatory network in the whole plant, but more importantly, at the root level. In response to abiotic stresses such as cold, heat, and salinity, their expression is significantly modified. To learn more about their transcriptional regulation, an integrative phenotypic, genomic, and transcriptomic study was executed on the roots of *A. thaliana* Pyrenean populations. Initially, the root phenotyping highlighted 3 Pyrenean populations to be tolerant to cold (Eaux), heat (Herr), and salt (Grip) stresses. Then, the RNA-seq analyses on these three populations, in addition to Col-0, displayed variations in *CIII Prxs* expression under stressful treatments and between different genotypes. Consequently, several *CIII Prxs* were particularly upregulated in the tolerant populations, suggesting novel and specific roles of these genes in plant tolerance against abiotic stresses.

## 1. Introduction

The haem peroxidases, such as the nonanimal peroxidase family, include three classes of peroxidases: Class I (CI Prxs), Class II (CII Prxs), and Class III (CIII Prxs) [1]. The CI Prxs are found in all kingdoms, the CII Prxs are secreted fungal lignin peroxidases, and CIII Prxs are plant-specific proteins. The latter belongs to a large multigenic family as a result of large gene duplication in vascular plants [2]. They are present in all Embryophytes but missing from Chlorophyceae [1]. They are apoplastic proteins characterized by dual hydroxylic and peroxidative cycles by which they can generate reactive oxygen species (ROS) and hence oxidize various substrates by using H_2_O_2_ as an electron acceptor, but they can also regulate the local concentration of H_2_O_2_ or produce ROS [3].

Therefore, they are involved in a myriad of physiological processes throughout the plant’s life cycle because of their genetic and catalytic diversities [4]. They contribute to biological processes such as auxin metabolism, lignin and suberin formation, and the defense against pathogens because of their antioxidative function. In addition, they regulate growth by maintaining a tight balance between cell wall loosening or de novo synthesis and stiffening [3].

Moreover, several CIII Prxs were characterized for their significant roles in providing tolerance to plants against abiotic stresses. For instance, *Prx62* was found to be implicated in salt tolerance during germination [5]. Additionally, the overexpression of subsets of *CIII Prxs* enhanced cold (*Prx22*, *Prx39*, and *Prx69*) [6] and heat tolerance (*Prx08*) [7]. Indeed, CIII Prxs expression and activity can be modified in accordance with external and internal cues to aid the plant in meeting its demands under stressful conditions [2].

The presence of 73 members in *A. thaliana* [8] suggests that each isoform is attributed a specific function. Their diversified promoters’ sequences support that their transcription is subjected to different regulatory pathways. Additionally, their protein sequences contain both conserved residues and domains such as those dedicated to heme binding, as well as highly variable domains certainly implicated in substrate binding and functional specificity [4]. The attempts to characterize their functional specificity are based on studying the spatiotemporal patterns of their expression and distribution [3]. For instance, they were detected all over the plant but more importantly in the roots [9], and the expression of 44 genes was localized and measured in different root tissues between different developmental stages [10].

With the increasing manifestations of climate change, plants are facing more frequent and severe abiotic stresses that drastically affect their growth and development [11,12,13]. Particularly, plant roots are prone to even greater damage than shoots as they are in direct contact with soils. In fact, plants can perceive the stress signals and produce appropriate responses with altered metabolism, growth, and development. Such responses are basically reliant on modifications in the expression of stress-related genes [14,15,16], including *CIII Prxs*. Notably, these transcriptional regulations are not linear circuits but are rather integrated networks involving multiple actors, where thousands of genes are differentially regulated under abiotic stresses [17].

*A. thaliana* has been adopted as a model species [18] because of its wide geographical distribution over Europe, Asia, and Africa, as well as its broad environmental distribution over distinct climates, which are the bases of its wide intraspecific diversity [19]. This study made use of the natural intraspecific variation present in *A. thaliana* Pyrenean ecotypes to analyze the transcriptional regulation of *CIII Prxs* in response to multiple abiotic stresses and in different populations. By definition, natural intraspecific variation is the inherent genetic diversity among individual organisms making up a population within a species. It originates from the accumulation and natural selection of spontaneous mutations that improve the fitness of their hosts to their direct environment [20]. It was extensively implemented in numerous studies to investigate the plants’ adaptations to their local environments [21,22,23,24,25]. Accordingly, it serves as an innovative and powerful tool to study the plants’ responses to abiotic stresses.

Therefore, this study implemented the natural intraspecific variation of 30 *A. thaliana* populations, distributed along an altitudinal gradient in the French Pyrenees, not covered yet in the 1001 Genomes Project [26], in addition to the well-studied ecotypes Columbia (Col-0, 200 m, Poland) and Shahdara (Sha, 3400 m, Tajikistan), to study the root development grown at 15, 22, or 28 °C with or without NaCl. Based on the observed root phenotypes, stress-tolerant populations were selected and subjected to transcriptional analyses to interpret the *CIII Prxs* regulation under multiple abiotic stresses. The objective of this study is to provide a detailed view of the roles of CIII Prxs in response to abiotic stresses in *A. thaliana* Pyrenean populations.

## 2. Results

The root development of the populations was studied under control conditions “ctrl” (22 °C) and under three stresses: cold stress “C” (16 °C), heat stress “H” (28 °C), and salt stress “S” (22 °C + 50 mM NaCl). Accordingly, three stress-tolerant populations were identified based on their root phenotypes under each treatment. Then, the seed germination of these selected populations was tested to mark the initiation of their root development. Afterward, transcriptional analyses were performed on Col-0 and the selected stress-tolerant populations to highlight the differential regulation of CIII Prxs in response to abiotic constraints.

### 2.1. Root Development Variation of Pyrenean Populations

The primary root growth of the multiple populations under different growth conditions was described by the root growth rates “RGRs” (mm/day) and the primary root lengths “PRLs” (mm). The aim was to highlight, on the one hand, the natural plasticity of the root development in the different *A. thaliana* populations, and on the other hand, the differential impacts of temperature and salt stress on this trait among the studied populations.

Overall, significant differences in the RGRs and PRLs were detected by two-way analyses of variance between the different treatments and the various populations, with *p*-values less than 0.0001 (Figure 1a). The mean of the root growth rates (RGRs) of the studied populations was accelerated at higher temperatures but decelerated due to cold and salinity. To illustrate, their median increased by 7% (2.24 mm/day) at 28 °C (H), but it was reduced by 63% (0.78 mm/day) at 16 °C (C) and by 47% (1.11 mm/day) due to salt stress (Figure 1b, Appendix A). Moreover, their PRLs medialized at 20.61 mm under ctrl conditions, and they were reduced by 50% (10.24 mm) under cold stress but slightly under heat stress by 3% (19.95 mm). Similarly, the salt treatment globally diminished the PRLs by 46% (11.15 mm) (Figure 1c, Appendix A). These results highlighted that the diverse abiotic stresses (cold, heat, and salinity) globally reduced the root lengths of the populations and were hence limiting their root growth.

The total root phenotypic data (RGRs and PRLs) of the populations under the four designed growth conditions were summarized using principal component analysis (PCA). The principal component 1 (PC1) explained 54% of the phenotypic variables, while PC2 explained 20%. The populations were differentially distributed along the PCA plot, and they mostly clustered apart from Col-0 and Sha, which highlighted their plasticity in response to abiotic stresses (Figure 1d). PC1 is mostly correlated with the ctrl treatment, whereas PC2 was positively correlated with the heat stress and negatively associated with the salt stress (Figure 1e). Consequently, populations such as Belc and Roch clustered near Col-0 grew best at 22 °C contrasting Mari and Biel. Eaux, Gedr, and Cast, which clustered in the middle, were envisaged to be cold-tolerant populations; however, Herr and Urdo were forecasted as heat-tolerant populations since they were biased towards positive PC2. These results showed that the Pyrenean populations responded differentially to the various treatments, which reflected their natural variation.

The development of the populations’ primary roots was tracked over 8 to 14 days, and it was contrasted in a genotype-dependent as well as treatment-dependent manner (Figure A1, Appendix A). For instance, Roch, Belc, and Bedo developed the longest roots (25.84, 24.51, and 24.18 mm) at the highest RGRs (2.64, 2.52, and 2.51 mm/day), while Mari, Pont, and Biel had the shortest ones (13.92, 15.43, and 15.67 mm) growing at the lowest rates (1.41, 1.60, and 1.60 mm/day) at 22 °C (Figure A2 and Figure A3, Appendix A).

At 16 °C, Eaux, Belc, and Roch developed the longest roots (13.98, 12.54, and 11.81) at the fastest rates (1.03, 0.97, and 0.88 mm/day), whereas Mari, Biel, and Bedo grew the shortest ones (7.56, 7.89, and 7.95 mm) most slowly (0.47, 0.61, and 0.62 mm/day). Interestingly, certain populations, such as Hosp and Mere, had moderate PRLs (10.14 and 10.34 mm) but with fast growth rates (0.88 and 0.86 mm/day). This phenomenon was due to the slow rates of seed germination of these populations, which delayed the root development initiation (Figure A2 and Figure A3, Appendix A).

To further highlight the impacts of the applied stress on the populations, the PRL of each population under each treatment was compared to that at 22 °C (ctrl). Under cold stress, the PRLs of all the populations decreased, yet the magnitudes of these reductions were differential according to the genotype. For instance, the PRLs of Eaux, Hern, and Lave were the least reduced at 16 °C compared to Arag, Argu, and Bedo (Figure 2a, Appendix A). Accordingly, Eaux was selected to study cold tolerance, as its root was minimally suppressed under cold stress compared with Col-0 (Figure 2b).

At 28 °C, the most elongated roots and fastest growth rates were observed in Urdo, Jaco, and Herr (PRLs: 26.07, 24.50, and 24.39 mm; RGRs: 3, 3.05, and 3.17 mm/day); however, Biel, Mari, and Hern had the shortest roots (13.49, 14.18, and 15.91 mm) with slowest RGRs (1.7, 1.71, and 2.03 mm/day). These observations revealed that the temperature differentially affected the root growth of the studied populations and that its effects were influenced by the different genotypes (populations) (Figure A2 and Figure A3, Appendix A).

Under heat stress, several populations developed longer roots compared with the ctrl, such as Herr, Urdo, and Jaco, while others developed shorter ones, such as Col-0, Roch, and Hosp (Figure 2c, Appendix A). Herr was accordingly selected as a good candidate to study heat tolerance, as its root growth was increased by heat, while Col-0’s PRL was reduced (Figure 2d).

Moreover, Hosp, Prad, and Guch developed the longest roots under salt stress (14.18, 13.86, and 13.56 mm), with higher growth rates for Hosp and Guch (1.46 and 1.40 mm/day) but a lower one for Prad (1.01 mm/day). Yet, the roots of Bier, Mere, and Mari were reduced because of salinity (PRLs: 7.54, 8.19, and 8.22 mm; RGRs: 0.82, 0.82, and 0.8 mm/day) (Figure A2 and Figure A3, Appendix A). Altogether, these results allowed classifying several populations as cold-, heat-, and salt-tolerant/sensitive according to their observed root lengths and growth rates under the different treatments.

Furthermore, the salt treatment globally decreased the PRLs and RGRs; however, these reductions were variable, depending on the population. To exemplify, the PRLs were moderately diminished in populations such as Savi, Prad, and Grip compared with other severely reduced populations such as Gedr, Bier, and Mere (Figure 2e, Appendix A). Correspondingly, Grip was picked as a good candidate to study salt tolerance as its root length was mildly reduced upon salt application compared with Col-0 (Figure 2f).

To sum up, the root phenotyping showed that the populations had varying root growth under control conditions which could be related to their genetic architecture. Additionally, their roots were generally reduced because of the applied treatments, except under heat for some populations. Most importantly, they unevenly responded to the applied stresses, where the root development of sensitive populations was more reduced compared with other tolerant ones. Accordingly, the root phenotyping allowed identifying some populations that had fairly advanced root growth under cold (Eaux), heat (Herr), and salt (Grip) stresses compared with Col-0. Then, the seed germination of the selected populations was tested to benchmark the beginning of their root development, aiming to know if they developed longer roots because they formerly germinated earlier or because they were truthfully tolerant to the stresses.

### 2.2. Seed Germination

Germination tests were performed on the cold-, salt-, and heat-tolerant populations (Eaux, Grip, and Herr, respectively) in addition to Col-0 under different combinations of treatments. These experiments highlighted the existing natural plasticity of this trait among the selected candidates and featured how cold, heat, and salt altered it. In addition, it was essential to assess the germination rates of the populations to normalize the effects of any significant variations that may advance or delay the initiation of root growth. For this aim, the testa rupture (TR) and the endosperm rupture (ER) were tracked over a time course (Appendix A). The germination percentage (GP in %) and the median germination time (t50 in hours) were computed (Appendix A) [27].

In Col-0, 100% of the seeds successfully germinated under the four conditions; however, the speed of their germination increased with the temperature but decreased with the salt (Figure 3a,b). To illustrate, the TR and ER t50s were 22.99 and 29.85 h at 22 °C (ctrl). Under cold stress (C), they increased, respectively, by 78% and 57%. However, under elevated temperature (H), they decreased by 29% and 33%. Furthermore, the salt treatment (S) significantly delayed germination, and the TR and ER t50s increased by 23% and 42% (Figure 3c,d).

These results suggested that the germination of Col-0 was fastest at 28 °C, but it decelerated at 16 °C and under salt stress. Additionally, the low temperature slowed the germination of Col-0 more than the salinity. Moreover, the cold stress reduced the TR more than the ER; the heat stress enhanced the TR and ER almost equally, whereas the salt stress decelerated the ER more than the TR.

The seeds of Eaux, the cold-tolerant population, fully germinated under the ctrl and cold conditions. At 22 °C, the TR and ER t50s recorded 43.46 and 55.92 h. They both increased under the cold treatment by 45% and 56%. Furthermore, the germination of Eaux was constantly slower than that of Col-0 under the two tested conditions. To illustrate, the TR t50s of Eaux were prolonged by 20.47 and 39.9 h compared with those of Col-0 at 22 °C and 16 °C, and its ER t50s were extended by 26.07 and 57.61 h under ctrl and cold conditions compared with Col-0 (Figure 3c,d).

Herr, the heat-tolerant population, entirely germinated at 22 °C and 28 °C. The TR and ER t50s recorded 51.67 and 65.03 h at the control conditions. The heat treatment (H) did not significantly decrease the TR t50 (6%), but it diminished the ER t50 by 16%. In addition, the germination of Herr was always lagging that of Col-0. For instance, the TR t50s in Herr were greater than those in Col-0 by 28.68 and 25.8 h at 22 °C and 28 °C, respectively, and its ER t50s were longer by 35.18 and 24.65 h under ctrl and heat conditions compared with Col-0 (Figure 3c,d).

Grip, the salt-tolerant population, completely germinated under the ctrl and salt conditions, although salinity delayed its germination. To illustrate, its TR and ER t50s were 41.13 and 35.29 h at 22 °C. Upon salt application, they increased by 20% and 65%, respectively. In reference to Col-0, Grip had slower germination under both conditions. To illustrate, the TR t50s were prolonged by 18.14 h and 26.16 h, and the ER t50s by 5.44 h and 28.24 h, respectively, under the control and salt stress (Figure 3c,d).

To sum up, the tests showed that each of the selected populations (Eaux, Grip, and Herr) had slower seed germination compared with Col-0 under both the control and the corresponding stress condition. However, the slowness in their germination was extended because of the different stresses, causing a subsequent delay in their root development initiation. However, despite these delays, the roots of Eaux, Grip, and Herr were less reduced compared with Col-0 under cold, salt, and heat treatments, respectively. Hence, their choice as convenient candidates for cold, salt, and heat tolerance was validated.

### 2.3. Omic Analysis

RNA-seq analyses were performed on Col-0 and the selected tolerant populations (Eaux, Herr, and Grip) under control conditions (22 °C) and under the corresponding stress for each, i.e., cold for Eaux, heat for Herr, and salt for Grip. The overall results were summarized in gene-level expression tracks (GEs) holding information about the expression values (TPMs) for each gene in the multiple populations grown under different conditions (Appendix A). The expression values (TPMs) of *CIII Prxs* were extracted (Appendix A).

The overall expression patterns of *CIII Prxs* under control conditions (22 °C) varied within and among the studied populations. For instance, a subset of five *CIII Prxs* was highly expressed in all populations (Figure 4a). However, other members of this multigenic family had higher expression either in Col-0 (Figure 4b) or in one of the Pyrenean stress-tolerant populations (Figure 4c–e).

Expectedly, these results showed that the expression of *CIII Prxs* was naturally variable among the studied populations. These variations could be attributed to variations in the cis-regulatory elements of their promoters’ regions. To support this hypothesis, the SNP frequency was analyzed from the DNA of each population. Regarding the whole CIII Prx family, 6920 variable positions have been detected between 2000 bp upstream and 1000 bp downstream, including 2415 in the upstream region. The number of SNPs is variable among the 73 CIII Prxs (from 30 to 256 for the *Prx28* and *Prx62*, respectively), with an average of 95 (Appendix A). It is less variable among populations, except for Hosp, with only 28 SNPs for the whole family. Eaux is the highest polymorphic population with 2012 SNPs, higher than other populations (average of 1600 SNPs).

In response to cold stress, a substantial number of *CIII prxs* were differentially expressed as part of the fundamental transcriptional modifications. While some of these regulations were observed in both Col-0 and the Pyrenean cold-tolerant population “Eaux”, others were exclusive to each genotype. In Col-0, only 2 *CIII prxs* (*Prx10* and *Prx71*) were significantly upregulated, while 15 genes were downregulated because of the cold treatment compared with the ctrl. However, nine members of this multigene family were upregulated in Eaux under cold, where only *Prx71* was mutual with Col-0. In addition, 10 genes were downregulated in Eaux under low temperature, where *Prx38*, *Prx52*, *Prx53*, *Prx54*, and *Prx65* were exclusively suppressed in this population. Comparing genotypes, 17 upregulated *CIII Prxs*, and 14 downregulated ones were present in Eaux versus Col-0 (Figure 5a).

The heat treatment triggered important changes in the expression of *CIII Prxs*, verifying that these genes were involved in response to elevated temperature. These regulations were variable between the populations, a fact which could be related to their genetic difference. They were more present in Col-0 rather than in the heat-tolerant population “Herr”, suggesting that the latter had a higher threshold of tolerance by which the applied stress could not trigger vast changes.

In Col-0, 11 *CIII Prxs* were significantly upregulated, and 11 were downregulated due to the heat treatment compared to the ctrl. However, only two genes were differentially expressed in Herr, where *Prx65* was upregulated, and *Prx21* was downregulated, similarly to Col-0. Comparing the two populations under heat stress showed that 6 *CIII Prxs* were upregulated, while 12 were downregulated in Herr (Figure 5b).

The salt stress also altered the expression of a myriad of genes, including *CIII Prxs*. In Col-0, 4 genes were significantly upregulated, while 12 others were downregulated because of the applied salinity to the growth media. In addition, three *CIII Prxs* were upregulated in the salt-tolerant Pyrenean population “Grip”, where *Prx49* was mutual with Col-0. Additionally, 10 genes were downregulated in Grip under salt stress, where *Prx12*, *Prx67*, and *Prx70* were exclusively inhibited in this population. Moreover, 11 upregulated *CIII Prxs* and 14 downregulated ones were detected by comparing Grip to Col-0 under salt conditions (Figure 5c).

### 2.4. Class III Peroxidase Activity

The total CIII Prx activity (PA) was specifically assayed using guaiacol/H_2_O_2_ [9] in Col-0 and in the selected stress-tolerant populations under the corresponding treatments. At 22 °C, the PA was 94 nkat/mg in Col-0. It was greatly enhanced by heat (168 nkat/mg) but insignificantly increased with cold (107 nkat/mg). Similarly, no significant difference in PA was observed under saline conditions (99 nkat/mg). Moreover, the PA in Eaux, the cold-tolerant populations, was 82 nkat/mg under ctrl conditions, insignificantly lower than that in Col-0 (Figure 6a). Unlike Col-0, it significantly increased because of the cold treatment (107 nkat/mg). Additionally, the PA, the heat-tolerant population Herr, was 80 nkat/mg at 22 °C, slightly less than that in Col-0. Surprisingly, it did not increase under heat conditions (85 nkat/mg) in contrast to Col-0 (Figure 6b). Furthermore, the PA in Grip recorded 119.85 nkat/mg at ctrl conditions; it did not significantly change due to the salt stress (109.7 nkat/mg) (Figure 6c).

These results showed that CIII Prxs had basal enzymatic activity in all studied populations under control conditions, proving that these proteins performed vital processes during cellular homeostasis. Their activity was importantly enhanced by heat in Col-0, which displayed their role in responding to temperature changes. Additionally, the low-temperature treatment in Eaux induced their activity, suggesting that they could contribute to cold tolerance. However, it did not increase in Herr, the heat-tolerant population, when exposed to elevated temperatures.

A correlation between the *CIII Prxs* expression and their peroxidase activity was noticed. To illustrate, a subset of 7 CIII Prxs was upregulated in Eaux, whereas only two of these genes were upregulated in Col-0 under the cold treatment. In addition, Eaux had more upregulated CIII Prxs at 16 °C when compared with Col-0 based on their genotypic difference. Coherently, this transcriptional pattern was reflected in the peroxidase activity, which was enhanced in Eaux under cold stress. Similarly, under hot conditions, 11 *CIII Prxs* were upregulated in Col-0, but only the expression of *Prx65* was enhanced in Herr. Moreover, the genotype-based DE analyses detected 12 upregulated genes in Col-0 versus Herr at 28 °C. Consistently, the peroxidase activity was triggered by elevated temperatures in Col-0 but not in Herr.

## 3. Discussion

### 3.1. The Phenotypic Variation of the Pyrenean Populations

On an evolutionary scale, plants adapted to their local environments by selecting genes that enhanced their fitness [28] and expanded their phenotypic plasticity. Consequently, they were enabled to optimize their morphologies and hence adapt to the requirements of their living habitats [29]. In fact, the local adaptation choreographed the genetic architecture of populations, benefitting from either the available genetic diversity or from novel variations generated by mutations or allele migration [30]. Indeed, the diversification of adaptive evolutionary trajectories between intraspecific populations was the basis of their natural variation [31]. Such adaptive mechanisms were broadly characterized in *A. thaliana*, where numerous genomic maps of climate adaptations were established for populations at macro and microgeographic scales [32].

The Pyrenean populations in this study filled a gap in the geographic distribution of *A. thaliana*, so they could serve as new tools to understand the genetic variation and the plant adaptations to abiotic stresses between the Iberian Peninsula and the rest of Europe. The particularity of this sampling was derived from their close geographical distribution but in highly contrasted environments due to the mountainous nature of their collection sites. To illustrate, these populations spanned an altitudinal gradient, and hence they were naturally exposed to variable climatic conditions such as annual minimum, maximum, and mean temperatures (°C), in addition to annual precipitation (mm) and total annual UV radiations (kWh/m^2^). Such variations would certainly trigger the evolutionary divergence among these populations.

In the Pyrenees Mountains, the local environmental conditions are highly contrasted, where temperature, precipitation, and UV radiation patterns are correlated with the altitude. In a previous study, the genetic structure of the populations was established, by which they were segregated into three clusters, and specific genetic lineages were detected among them. This genetic diversity was translated into contrasted phenotypes of these populations when exposed to suboptimal temperatures [33].

In this study, the populations’ roots phenotypes were also contrasted between different abiotic stresses. The root lengths and growth rates were used as indices to estimate the root development in the populations. In fact, roots play fundamental roles in the plant by foraging underground resources such as minerals and water. They are also known for their enormous plasticity by which they can respond to extrinsic cues via modulating their growth [34]. Therefore, proper root development features healthy plants, but when root growth is tackled by abiotic constraints, the whole plant development is constrained.

To sum up, differential root lengths and growth rates were observed between different treatments and various populations. The multivariate analyses of the root phenotypic data allowed the classification of the populations into cold-, heat-, and salt-tolerant or sensitive. For instance, Arag and Bedo were cold-sensitive populations since they developed tiny roots at 16 °C, but Eaux and Lave were cold-tolerant populations having relatively longer and faster-growing roots under cold conditions. These cold-sensitive and tolerant populations were formerly characterized to accumulate low and high anthocyanin content, i.e., stress indicator, in their rosettes under cold stress (Duruflé et al. 2019 [33]).

Moreover, the seed germination of Col-0 and selected stress-tolerant populations (Eaux, Herr, and Grip) was studied under control and stressful conditions, which determined their germination timing and hence the beginning of their root development. This experiment enabled distinguishing whether the stress-tolerant populations developed longer roots because of early germination and growth initiation or because of their potential to develop properly under stress.

In fact, seed germination is a crucial event that marks the beginning of the plant’s life cycle and ensures its survival [35]. It is regulated by both internal (phytohormones such as gibberellin and abscisic acids) and external signals (temperature, precipitation, and light spectrum) [36,37]. Different abiotic stresses can restrain the proper germination of seeds, which may impact the early seedling establishment and even later processes, such as delaying the bolting stage [38]. Relative to the environmental conditions, seeds may germinate faster so the plant can grow under preferential conditions, or they may slow their germination pending future favorable conditions [39].

The seeds of Col-0, Eaux, Grip, and Herr fully germinated under control and under stressful conditions, but their germination speed varied between genotypes and treatments. For instance, Col-0 had the fastest germination rate compared with the selected Pyrenean populations under all tested conditions. However, the selected populations developed longer roots under stressful conditions compared with Col-0 despite the delay in their growth initiation caused by the delay in their seed germination. This observation validated that these populations were stress tolerant regarding their root development since they developed longer roots faster compared with Col-0. Indeed, these observations highlighted the phenotypic plasticity among the Pyrenean populations, which could be related to their genetic diversity.

### 3.2. CIII Prxs Transcriptional Regulation in the Pyrenean Populations

RNA-seq was adopted as a high-throughput technique to study the whole transcriptome of an organism at once [40]. Transcriptomics assesses the activity of genes in different biotic and abiotic contexts by measuring their expression values. Consequently, genes are associated with different physiological roles, such as responding to environmental stresses [41,42]. In parallel, the allelic frequency for the various population was analyzed in reference to the Col-0 genome for the 73 CIII Prxs. The number of SNPs is highly variable among the genes, with one-third of the SNP found in the upstream regions, which can explain the differential expression of the genes depending on the growth conditions and the populations. The number of SNPs is not highly different among the population except for Hosp, which is in agreement with the previous study made with few sequences [33].

This study focused on the expression of the *CIII Prxs* in three stress-tolerant Pyrenean populations, in addition to Col-0, under control and stressful conditions. In fact, *CIII Prxs* is a larger gene family including a plurality of members compared with ascorbate peroxidase (APx, member of CI Prx) and glutathione peroxidase (GPx, part of the ROS gene network together with CIII Prx, APx, and multiple other protein families) [43]. In addition, *CIII Prxs* were previously characterized as extensively involved in many physiological processes and play key roles during the plant’s response to environmental constraints [44], during which their expression can be regulated as part of this response [2].

As for *CI Prxs*, only a few genes were found significantly differentially expressed in the Pyrenean populations. To exemplify, the treatment-based DE analyses showed that *GPx02* and *GPxO6* were upregulated in Eaux under cold stress compared with the optimal condition, and *APx01* was enhanced in Grip under salinity. Yet, none were triggered in Herr when exposed to heat. Additionally, the genotype-based analyses highlighted other *CI Prxs* with higher expression in the Pyrenean populations compared with Col-0, such as *APx01* and *GPx07* in Eaux at low temperature, *APx01* and *APx04* in Herr at high temperature, and *APx01* in Grip at saline conditions. Despite such significant changes in the expression of *APxs* and *GPxs* between genotypes and in response to abiotic stresses, the variations detected in *CIII Prxs* were more multitudinous.

Initially, *CIII prxs* transcriptomic data under control conditions vividly demonstrated the natural variation in their expression between the Pyrenean populations, which could be attributed to their underlying genetic diversity. Moreover, the treatment-based differential expression (DE) analyses showed that the expression of many *CIII Prxs* was altered in response to the applied stress in each population. Additionally, the genotype-based DE analyses displayed that their regulation was variable between the stress-sensitive Col-0 and the stress-tolerant Pyrenean populations under each condition. Consequently, their contrasted expression between treatments and genotypes allowed the characterization and specification of novel roles of these genes in providing tolerance to plants.

The treatment-based differential expression (DE) analyses highlighted subsets of two and nine upregulated *CIII Prxs* in Col-0 and Eaux, with *Prx71* mutually upregulated in both populations. These results suggested novel roles in cold tolerance of the Eaux-exclusive eight upregulated genes (*Prx07*, *Prx08*, *Prx44*, *Prx55*, *Prx56*, *Prx57*, *Prx60*, and *Prx73*). Furthermore, another subset of 17 upregulated *CIII Prxs* in Eaux compared with Col-0 was identified by the genotype-based DE analyses, suggesting putative roles of 9 additional genes to be involved in cold tolerance (*Prx01*, *Prx16*, *Prx24*, Prx27, *Prx37*, *Prx47*, *Prx64*, *Prx66*, and *Prx70*) (Figure 7a).

The treatment-based DE analyses identified subsets of 11 and 1 upregulated *CIII Prxs* in Col-0 and Herr, with *Prx65* mutually upregulated in both populations, signifying that these genes were involved in heat tolerance (*Prx01*, *Prx08*, *Prx17*, *Prx35*, *Prx39*, *Prx50*, *Prx53*, *Prx58*, *Prx60*, *Prx65*, and *Prx73*). Additionally, the genotype-based DE analyses, which highlighted a subset of six *CIII Prxs* to be upregulated in the heat-tolerant population “Herr”, suggested additional putative roles of these genes in response to elevated temperatures (*Prx25*, *Prx33*, *Prx37*, *Prx61*, *Prx67*, and *Prx71*) (Figure 7a).

The treatment-dependent DE analyses identified subsets of four and three upregulated *CIII Prxs* in Col-0 and Grip, with *Prx49* mutually upregulated in both populations, proposing new functions of the Grip-exclusive upregulated genes in the response against salty conditions (*Prx15* and *Prx62*). Furthermore, the genotype-dependent DE analyses, which exposed a subset of 11 upregulated *CIII Prxs* in the salt-tolerant population “Grip” compared with Col-0, suggested additional putative roles of these genes in response to salinity (*Prx07*, *Prx08*, *Prx14*, *Prx25*, *Prx33*, *Prx35*, *Prx40*, *Prx44*, *Prx56*, *Prx71*, and *Prx73*) (Figure 7a).

Compiling these results allowed a functional specification of temperature- and salt-related *CIII Prxs*. For instance, some *CIII Prxs* were significantly upregulated in the cold-tolerant population Eaux compared with Col-0 under low temperature (*Prx16*, *Prx24*, *Prx27*, *Prx47*, *Prx64*, *Prx66*, and *Prx70*) or in Eaux at 16 °C compared with 22 °C (*Prx55* and *Prx57*), suggesting that they were particularly involved in cold tolerance. These genes were previously characterized for other roles such as Prx16, which was involved in seed germination [5], Prx24 in auxin and brassinosteroid signaling [45], Prx27 in maintaining postembryonic root growth [46], Prx47 in lignification [47], Prx64 in Casparian strip lignification and SGN-dependent compensatory lignification [48], Prx66 in lignifying the tracheary root elements [47], and Prx70 in seed germination [49]. Additionally, Prx55 was linked to pathogen response and cell cycle regulation during geminiviral infection [50] and Prx57 to the increased accumulation of ROS and permeability of leaf cuticle [51] (Figure 7b).

However, genes that were upregulated in the heat-tolerant population Herr compared with Col-0 under high temperature (*Prx61* and *Prx67*), or in either Col-0 or Herr at 28 °C compared with 22 °C (*Prx17*, *Prx39*, *Prx53*, *Prx58*, and *Prx65*) were associated to the response against heat stress. Formerly, Prx61 and Prx67 were not associated with any functions in any context. However, Prx17 was a direct target of the AGAMOUS-LIKE15 transcription factor and contributed to lignification [52]. In addition, Prx39 was involved in lignification at the level of the Casparian strip [48], Prx53 in stem lignification [53], Prx58 in pollen tube polar growth [54], and Prx65 in pollen germination and tube growth [55] (Figure 7b).

Nevertheless, other *CIII Prxs* which were upregulated in the salt-tolerant population Grip compared with Col-0 under saline conditions (*Prx14* and *Prx40*), in Grip at salty conditions compared with ctrl conditions (*Prx15* and *Prx62*), or in Col-0 (*Prx49* and *Prx52*) had specific roles in responding against salinity. The corresponding proteins were formerly known for playing distinct roles in plants, except for Prx14. For instance, Prx40 was found essential to proper anther and pollen development [56]. As for the Grip-exclusive upregulated genes under salt stress, Prx15 was involved in ROS-induced programmed cell death [57], while Prx62 was related to seed germination as it was expressed in the micropylar endosperm [5], and it was recently characterized to be also involved in regulating root hair elongation under cold conditions [58]. Moreover, Prx49 and Prx52, which were upregulated in Col-0 under salt stress, were involved in the abscisic acid early signaling in Arabidopsis [59] and in lignin biosynthesis and xylem development [60], respectively (Figure 7b).

In addition, some genes were upregulated under different treatments in the Pyrenean populations suggesting that they were involved in numerous stress-induced physiological responses. To exemplify, *Prx01*, *Prx37*, and *Prx60* had higher expression under both cold and hot temperatures, and they hence played roles in responding to temperature variations. Formerly, Prx01 was identified as an actor in pathogenesis, cold tolerance, and extension regulation during root hair growth [61], Prx37 in the resistance to *Botrytis cinerea* [62], and Prx60 in the polar growth in the pollen tubes [54] and the trichoblasts of the root epidermis [63] (Figure 7b).

Furthermore, *Prx07*, *Prx10*, *Prx44*, and *Prx56* were associated with both cold and salt stresses. *Prx07* was previously found as a putative target of the ROOT HAIR DEFECTIVE SIX-LIKE4 (RSL4) gene regulating root hair development [64]. Prx10 was known for its role in controlling root elongation (Markakis et al., 2012). Prx44 was linked to the response to salinity [65], while Prx56 was related to mucilage extrusion in the seed coat [66] (Figure 7b).

Similarly, *Prx25*, *Prx33*, *Prx35*, and *Prx50* were enhanced by both heat and salt stresses. Prx25 was characterized as an actor in cell wall lignification and seed longevity [67,68], while Prx33 acted in response to photoperiod stress [69]. Additionally, *Prx35* was highly expressed in the trichoblasts and was hence associated with root hair formation [63]. Prx50 was involved in pathogen response and cell cycle regulation during geminiviral infection [50] (Figure 7b).

Finally, a subset of three *CIII Prxs* (*Prx08*, *Prx71*, and *Prx73*) was triggered by all applied stresses, i.e., cold, heat, and salinity. These genes were previously identified in other contexts, such as Prx08, which was involved in regulating epidermal differentiation in Arabidopsis roots [63]. Moreover, Prx71 contributed to the lignification of the secondary cell walls [70], while Prx73 was detected during root epidermal differentiation [63] (Figure 7b).

## 4. Materials and Methods

### 4.1. Biological Material

Thirty new-found *A. thaliana* Pyrenean populations were studied in addition to two external ecotypes originating from contrasted altitudes, Columbia (Col-0, 200 m, Poland) and Shahdara (Sha, 3400 m, Tajikistan). Plant individuals of these populations were previously gathered from different locations in the French Pyrenees, the mountainous physical barrier separating the Iberian Peninsula from France. Their names, climatic data, and taxonomic relevance to *A. thaliana* were previously reported [33]. To minimize the impacts of maternal effect, seeds were amplified at once under controlled conditions to obtain homogenous batches of seeds prior to further experiments.

### 4.2. Root Phenotyping

The primary roots lengths of the 32 studied populations were measured over a timeline under four growth conditions. Plants were grown in vitro in vertical positions inside 12 × 12 square Petri dishes at four contrasted conditions on half Murashige and Skoog basal medium (½ MS) with sucrose and agar (M9274-10L, Sigma-Aldrich, Saint-Quentin-Fallavier, France) and with the addition of 50 mM NaCl (GUANAC0166, Eurobio, Les Ullis, France) when salt stress was applied. The seeds were sterilized with a bleach solution (69.9% water, 30% Javel water (9.6% Cl), and 0.1% Triton), followed by three successive washes with autoclaved water. For each biological replicate, 16 to 18 seeds per population were sown along two lanes with even spacing; each lane contained 8 seeds from 2 populations to maximize homogeneity and randomness. Afterward, a cold stratification treatment was performed at 4 °C for 24 h in the dark to synchronize and promote germination before transfer to long day (16 h light/8 h dark) growth chambers (light intensity = 150 µmol/m^2^·s^−1^).

The plants were grown at three different temperatures designating a control “Ctrl” (22 °C), cold stress “C” (16 °C), and heat stress “H” (28 °C). They were grown with 50 mM NaCl at 22 °C (salt stress: “S”). The plants were scanned on a flat-bed scanner (Expression 12000XL, Epson, Levallois-Perret, France) at 1200 dpi over a timeline adapted for each temperature. The scanning time points were 0, 3, 6, 8, 10, and 14 days at 16 °C; 0, 2, 3, 4, 6, 8, and 10 days at 22 °C; and 0, 2, 3, 4, 6, and 8 days at 28 °C.

The primary roots were measured using the NeuronJ plug-in in ImageJ. The experiments were carried out in 3 to 12 replicates. Most data analyses were performed using the R software (Version 1.2.1335). First, the root growth rates RGRs (mm/day) were computed as the slopes of the linear regression lines of the primary root lengths (PRLs) measurements (mm) as a function of time (days). Second, the arithmetic means and the standard errors of means (SEM = SD/ N where SD is the standard deviation and N is the sample size of the PRLs were calculated for the various populations at the subsequent time points (days). Then, the differences among the PRLs recorded on the last day of growth were measured in percentage to evaluate their variation in each population and under each treatment in reference to the control (22 °C). Additionally, the variations in the PRLs and the RGRs due to the different treatments in the populations were statistically tested using the analysis of variance, two-way ANOVA, from the “multcomp” R package [71].

### 4.3. Germination Test

Germination tests were performed on seeds of three selected cold-, heat-, and salt-tolerant populations (Eaux, Herr, and Grip, respectively), in addition to Col-0, to estimate the effects of the corresponding treatments and different genotypes on seed germination. The germinating seeds were quantified up to 144 h. The three stages of germination were no rupture of the seed envelope (NR), rupture of the external testa envelope (TR), and rupture of the internal endosperm envelope (ER) marked by the embryo’s radicle protrusion. The statistical analyses were performed using the “germinationmetrics” R package [72].

The germination tests were performed in vitro inside 12 × 12 square Petri dishes on ½ MS +/− 50 mM NaCl. At least 100 sterilized seeds of each population were put on the media surface, stratified for 48 h at 4 °C, and then placed in a growth chamber (16 h light/8 h dark, light intensity = 150 µmol/m^2^·s^−1^). Col-0 germination was tested under all conditions, whereas each of the three selected Pyrenean populations was tested under the control (22 °C) and the stress condition to which it was tolerant. The quantification was performed using a binocular loupe (Zoom Pro 10.25, Perfex Sciences, Escalquens, France). The experiments were conducted in triplicates.

### 4.4. Transcriptomic Analyses

#### 4.4.1. Samples Preparation and Sequencing

The transcriptomic analyses were performed on the reference population Col-0 grown at the four designed growth conditions: the cold-tolerant population Eaux grown at the control conditions (22 °C) and under cold stress “C” (16 °C), the heat-tolerant population Herr grown at the ctrl and under heat stress “H” (28 °C), and finally the salt-tolerant population Grip grown at the ctrl and under salt stress “S”. The roots were harvested into microcentrifuge tubes containing metallic beads after 14, 10, or 8 days for plants grown at 16 °C, 22 °C, and 28 °C, respectively, and immediately frozen in liquid nitrogen.

The plants were ground via RETSCH Mixer Mill MM 400 for 2 min at 30 Hertz. The RNA was extracted using the Qiagen RNeasy Plant Mini kit according to the manufacturer’s instructions and dissolved into 50 μL elution buffer. The yield of RNA was determined by taking spectrophotometric readings by a Denovix DS-11 Nanodrop spectrophotometer. RNA sequencing and the preparation of RNA libraries were performed at Genewiz in Leipzig, Germany.

#### 4.4.2. Bioinformatic Analyses

The obtained data were treated using Qiagen CLC Genomics Workbench software (Version 21.0.2, Qiagen, Cortaboeuf, France). First, the sequenced reads were trimmed prior to mapping using quality-scores-based trimming (quality limit threshold = 0.5), trimming of ambiguous nucleotides (maximum number of ambiguities = 2), sequencing adaptors trimming (Read 1: AGATCGGAAGAGCACACGTCTGAACTCCAGTCAC; Read 2: AGATCGGAAGAGCGTCGTGTAGGGAAAGAGTGT), and length trimming by which short reads less than 50 nucleotides were discarded. Then, the trimmed reads were mapped against the *A. thaliana* cDNA library (TAIR10_cdna_20101214_updated) in batches, one reference sequence per transcript. The mapping process was parameterized by a match score = 1, a mismatch score = 2, a linear gap cost with insertion and deletion costs = 3, length fraction = 0.95, a similarity fraction = 0.98 to maximize stringency, and a maximum number of hits per read = 10.

TPM or “Transcripts per million”, a normalization method for RNA-seq, was selected as the expression value (TPM = $\frac{\mathrm{RPKM}\;\times \;10^{6}}{\sum^{\;}\mathrm{RPKM}}$) to normalize the sequencing depth and gene length within the samples. The RPKM or “Reads Per Kilobase of transcript per Million mapped reads” scales by the gene length, assuming that longer transcripts would generate more sequencing reads, and it is hence computed accordingly (RPKM = $\frac{\mathrm{total}\;\mathrm{exon}\;\mathrm{reads}}{\mathrm{mapped}\;\mathrm{reads}\;\left(\mathrm{millions}\right)\;\times \;\mathrm{exon}\;\mathrm{length}\;\left(\mathrm{KB}\right)}$). The outcome of the RNA-seq analyses was presented by gene-level expression tracks (GEs) holding information about the read counts and expression values for each gene in the different populations under the various treatments.

The analyses of the genes differential expression (DE) were performed on the whole transcriptome using statistical differential expression tests for the set of expression tracks with the associated metadata by multifactorial statistics based on a negative binomial linear model (GLM) while assuming that the read counts follow a negative binomial distribution [73]. The effects of both the treatment and the genotype were tested for the DE analyses, as both factors contribute to the differential expression of genes between the populations under the different treatments. A threshold of absolute fold change (FC) greater than or equal to 1.5 and a false discovery rate *p*-value (FDR) less than or equal to 0.05 was used to detect the significant differentially expressed genes (DEGs).

### 4.5. Genomic Analysis

Genomic DNA was extracted from a pool of individuals for each population. The rosette leaves of 3- to 4-week-old plants were harvested and frozen in collection microtubes (19560, Qiagen) containing metallic beads. They were then ground for 2 min at 30 Hertz using the RETSCH Mixer Mill MM 400. The obtained powder in each well was incubated with 400 μL of a DNA extraction buffer 0.5 M NaCl, 100 mM Tris, 50 mM EDTA, 1.25% sodium dodecyl sulfate (SDS) (EU0660, Euromedex, Souffelweyersheim, France), 1% PVP, 1% sodium metabisulfite (S9000, Sigma-Aldrich), 10 μg/mL RNase A (A7973, Promega, Charbonnières-les-Bains, France), and 0.5 mg/mL proteinase K (V3021, Promega)) at 65 °C for 30 min. After cooling and centrifugation, 130 μL of 4 M potassium acetate (P1190, Sigma-Aldrich) and 0.35 M acetic glacial acid (64-19-7, Thermo Fisher Scientific, Illkirch-Graffenstaden, France) mixture were added and the plate was incubated at −20 °C for 30 min. The resulting supernatants were transferred into a 96-well column plate (016086, Dutscher, Bernolsheim, France), where 600 μL of the fixation buffer (8 M guanidine chloride (50937, Sigma-Aldrich) and 100% ethanol (64-17-5, Thermo Fisher Scientific)) were added to each sample. After centrifugation, the new supernatants were transferred into a 96-well AcroPrepTM Advance filter plate (# VWR 738-0120, Pall Laboratory, Portsmouth, United Kingdom) fixed above a new 96-well column plate. The plates’ block was centrifuged, and the effluents were discarded. The filters were washed twice with 650 μL of a washing buffer (1 M Tris, 0.5 M EDTA, 5 M potassium acetate, and 100% ethanol). Finally, DNA was eluted into an elution plate (016061, Dutscher) with 50 μL of Tris-EDTA buffer added to each well. The extracted genomic DNA from 1 to 5 individuals corresponding to each population were pooled together into equimolar DNA pools that were sent for Illuminas sequencing by NovaSeq 6000 sequencer at the GeT-PlaGe GenoToul platform. The raw fastq reads were cleaned by removing the adapters and the low-quality sequences using cutadapt (v2.1) [74] and TrimGalore! (v0.6.5, https://github.com/FelixKrueger/TrimGalore) with -q 30 --length 20 options. The cleaned reads were mapped onto the TAIR10 A. thaliana reference genome Col-0 using bowtie2 (v 2.3.5.1) [75] with -no-discordant, -no-mixed options. Duplicated reads were removed using SAMtools (v1.9) [76] markdup command. A semistringent SNPCalling across the genome was performed for each accession with SAMtools mpileup and VarScan (v.2.4.2), VarScan mpileup2snp –min-coverage 8 –min-reads2 4 –min-avg-qual 30 –*p*-value 0.01 –strand-filter 1 –output-vcf 1. All polymorphic sites were then identified among the populations. Finally, an SNP calling based on all accessions was performed on all polymorphic sites to differentiate null values (NA) from the reference allele. The single nucleotide polymorphisms (SNPs) identified were then located on the Col-0 genome and have been classified as (i) upstream, (ii) downstream, (iii) in the coding, or (iv) in the noncoding regions of each gene. In the frame of this study focused on CIII Prxs, only SNPs variation in those genes has been presented.

### 4.6. Peroxidase Activity

The peroxidase activity was assayed using guaiacol/H_2_O_2_. The total proteins from selected populations grown under contrasting conditions were extracted with 10 mM Tris Buffer (EU0011, Euromedex), containing 20 mM EDTA (EU0007, Euromedex), and 5% PVP (PVP40-50G, Sigma-Aldrich). The protein concentrations were measured following Bradford’s protein quantification assay in 96-well plates (260860, Thermo Fisher Scientific). After the addition of the diluted Bradford reagent (1X) (39222, SERVA, Heidelberg, Germany), the optical densities of the samples were recorded at 590 nm wavelength. Formerly, the spectrophotometer (BioTek Elx 808, Thermo Fisher Scientificwas calibrated using increasing quantities from 1 to 5 µg of bovine serum albumin (BSA, 1005-70, Euromedex). To estimate the peroxidase activity, an 8:1 ratio of 200 mM phosphate buffer pH = 6 containing 0.125% guaiacol (50880, Sigma-Aldrich) was added to 11 mM H_2_O_2_ (412072, CARLO ERBA) and incubated immediately with the protein extracts. OD470 was successively registered 1 and 2 min after the reaction started. The peroxidase-specific activity was calculated (nanokatal/mg proteins). Two-way ANOVA tests for multiple independent samples were carried out to determine the effects of the different treatments and the populations on the peroxidase activity [71].

## 5. Conclusions

This study benefitted from the phenotypic plasticity displayed by *A. thaliana* Pyrenean populations that developed contrasted root phenotypes under thermal and saline stresses. This contrast allowed the identification of three stress-tolerant populations from the Pyrenean collection—Eaux against cold, Herr against heat, and Grip against salinity.

Afterward, the transcriptional regulation of *CIII Prxs* in these stress-tolerant populations was elaborately analyzed via RNA-seq. As expected, a myriad of genes had different expressions among populations under control conditions, reflecting their genetic diversity. Additionally, the expression of these genes significantly changed because of stressful treatments. Interestingly, these changes were contrasted between the Pyrenean populations and Col-0.

The treatment- and genotype-based differential expression (DE) analyses revealed additional aspects of the *CIII Prxs* regulations. To illustrate, several *CIII Prxs* were exclusively upregulated in the Pyrenean populations under stressful conditions, and hence novel roles in stress tolerance were associated with these genes. Furthermore, the compilation of these comparisons upgraded the available knowledge about the functional specificity of different *CIII Prxs*, where particular genes were identified to be involved in response to specific stress.

Indeed, the novel and specific putative roles in stress tolerance that were associated with cIII genes opened future perspectives to study tolerance mechanisms. Since such mechanisms rely on complex and interconnected regulatory pathways, further studies should integrate all biological aspects and scales from the molecule to the organism to achieve a better understanding of plant tolerance.

## Figures and Tables

**Figure 1 ijms-23-03960-f001:**
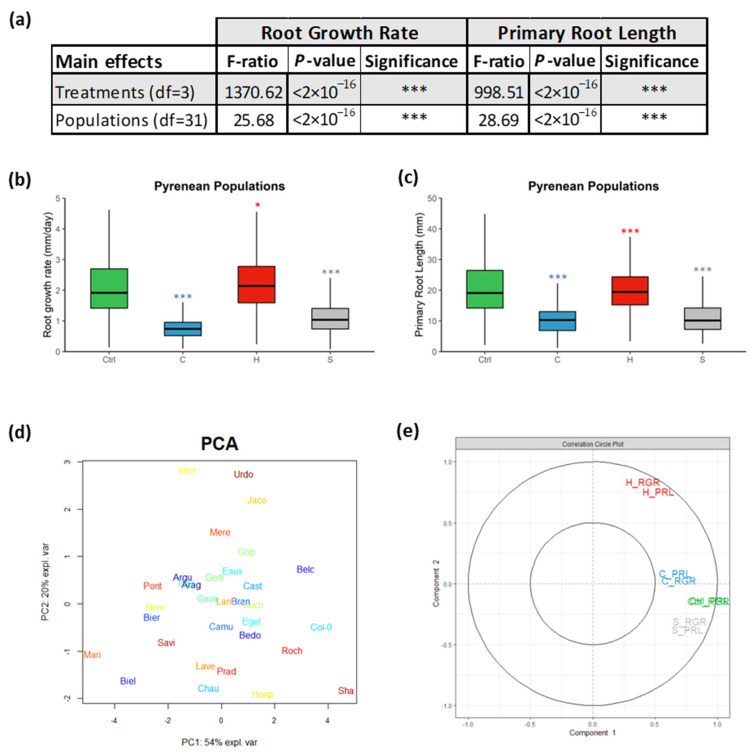
(**a**) Two-way ANOVA showing the main effects of the “Treatments” and the “Populations” on the two measured root parameters: “Root Growth Rate” and “Primary Root Length”. Boxplots displaying (**b**) the root growth rates (mm/day) and (**c**) the primary root lengths (mm) of the *A. thaliana* Pyrenean populations. PCA score plot (**d**) displaying the studied populations spanning the principal components PC1 (54%) and PC2 (20%) that summarize the measured primary root lengths (PRLs) and root growth rates (RGRs) under the four treatments. Correlation circle (**e**) representing the contribution of the measured variables to the principal components (PC1 and PC2). (***) signified the *p*-value < 0.0001 of the Student’s *t*-test of the pairwise comparisons between each treatment and the ctrl on the PRL and RGR of each population. * *p*-value < 0.01.

**Figure 2 ijms-23-03960-f002:**
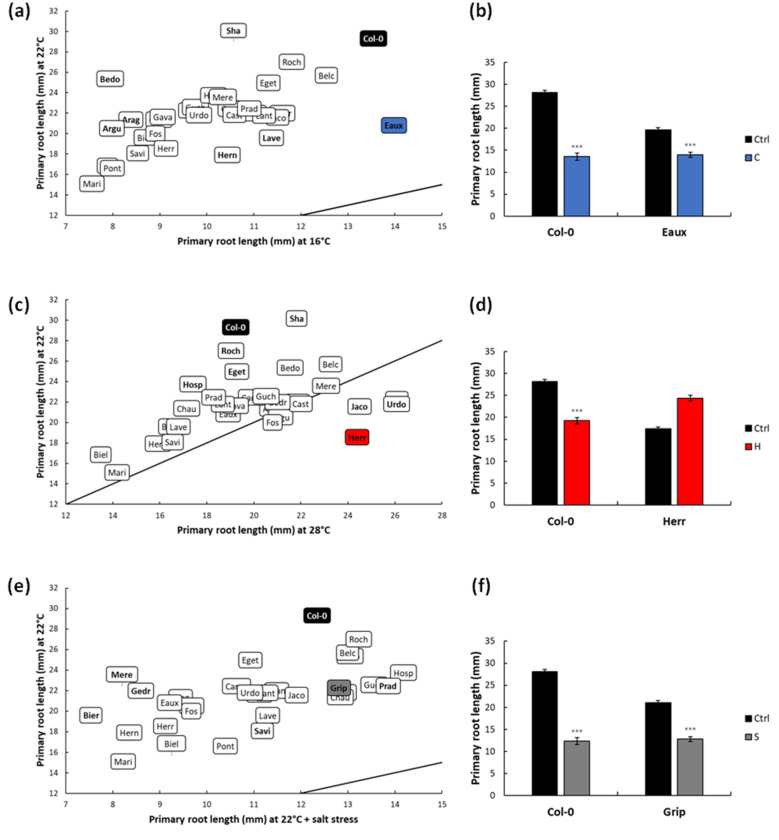
Scatter plot displaying the distribution of the populations according to their primary root lengths (PRLs) under ctrl and under (**a**) cold, (**c**) heat, and (**e**) salt stresses. The selected tolerant population under each successive treatment was highlighted blue, red, and grey, while Col-0 was highlighted in black. The line designates (y = x). Bar graphs displaying (**b**) the PRLs of Col-0 and Eaux under ctrl and cold conditions, (**d**) the PRLs of Col-0 and Herr under ctrl and heat conditions, and (**f**) the PRLs of Col-0 and Grip under ctrl and salt conditions. (***) signified the *p*-value < 0.0001 of the Student’s *t*-test of the pairwise comparisons between each treatment and the ctrl on the PRL of each population. The error bars represented the standard error of means (SEM).

**Figure 3 ijms-23-03960-f003:**
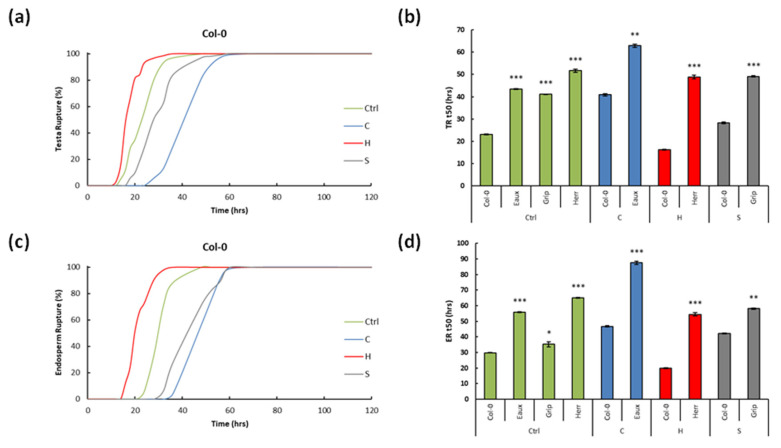
Line graphs showing (**a**) the testa and (**b**) the endosperm ruptures over time (h) in Col-0 under the four tested conditions; bar graphs revealing the median germination time t50 (h) of (**c**) the TR and (**d**) ER of Col-0 and the selected Pyrenean populations under the designed treatments: Col-0 and Eaux under C, Col-0 and Herr under H, and Col-0 and Grip under S. (***) *p*-value < 0.0001, (**) *p*-value < 0.001, and (*) *p*-value < 0.01 of the Student’s *t*-test of the pairwise comparisons between Col-0 and the Pyrenean populations at each treatment. The error bars correspond to the standard error of means (SEM).

**Figure 4 ijms-23-03960-f004:**
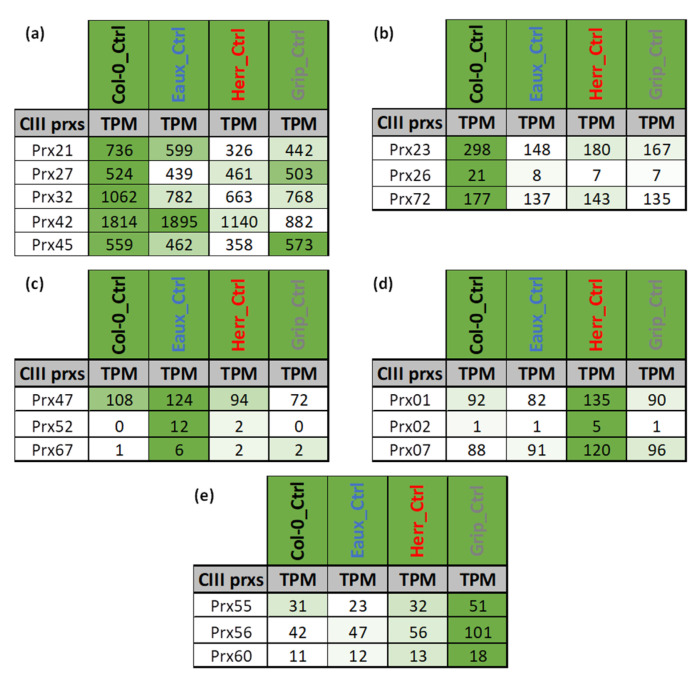
Tables of the averaged gene expression levels in transcripts per million (TPMs) in Col-0 and three Pyrenean populations (Eaux, Herr, and Grip) under control conditions (22 °C): (**a**) Subset of CIII Prxs with high expression in all populations; (**b**) CIII Prxs with higher expression in Col-0, (**c**) in Eaux, (**d**) in Herr, and (**e**) in Grip. The color code scales the expression level of each gene across the samples (white < green).

**Figure 5 ijms-23-03960-f005:**
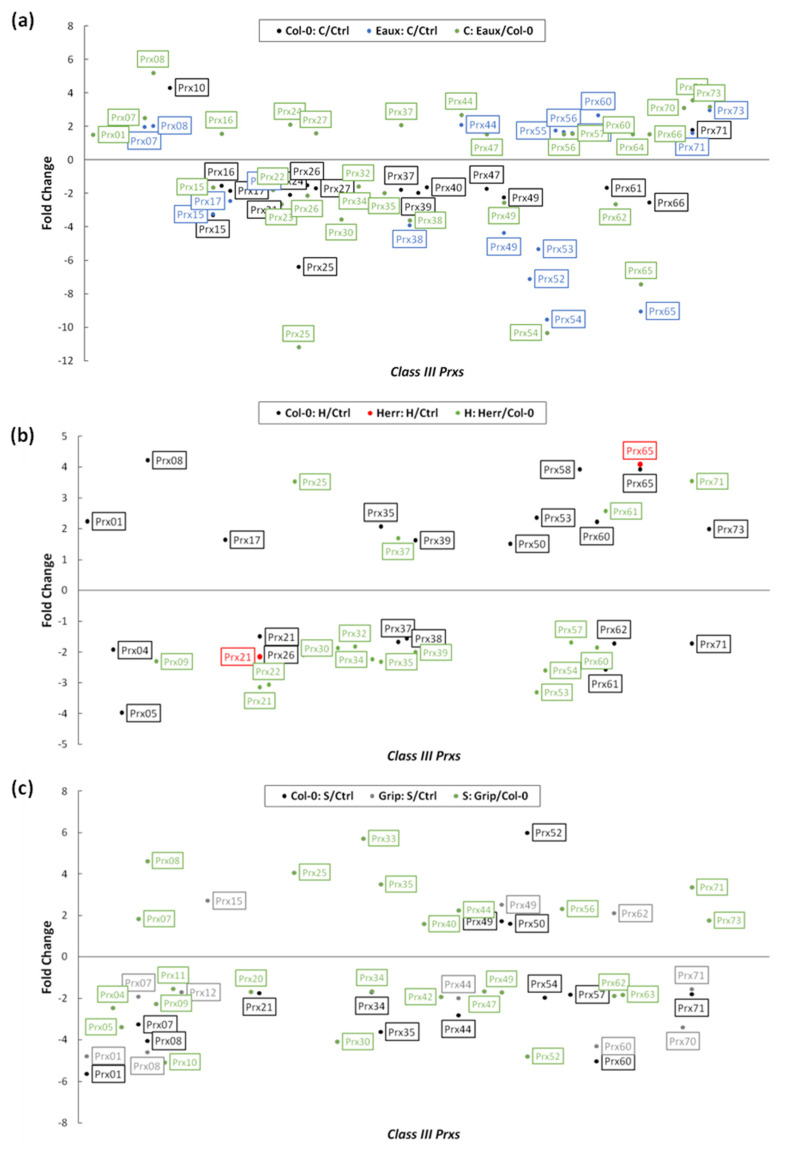
Scatter plots representing the significant differential expression of *CIII Prxs* (fold change |FC| ≥ 1.5 and false discovery rate FDR ≤ 0.05) in (**a**) Col-0 between cold and control conditions (black), Eaux between cold and control conditions (blue), and in cold conditions between Eaux and Col-0 (green); (**b**) Col-0 between hot and control conditions (black), Herr between hot and control conditions (red), and in hot conditions between Herr and Col-0 (green); (**c**) Col-0 between salt and control conditions (black), Grip between salt and control conditions (grey), and in salt conditions between Grip and Col-0 (green).

**Figure 6 ijms-23-03960-f006:**
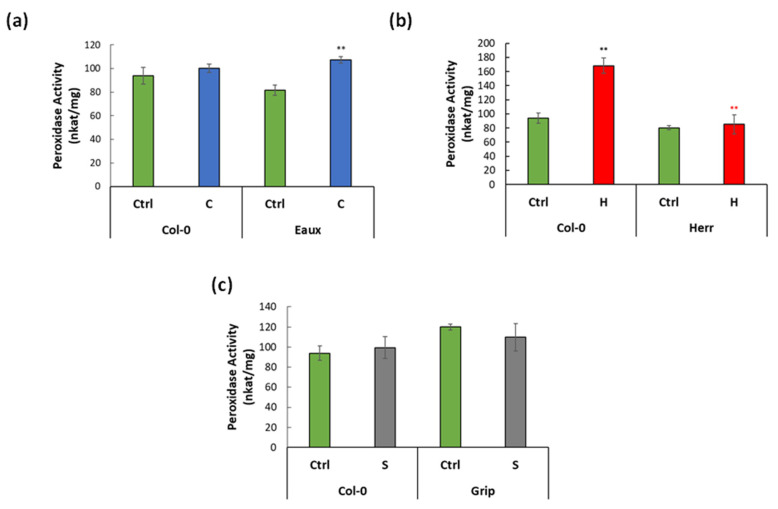
Bar plots displaying CIII Prx activity (nkat/mg of proteins) in (**a**) Col-0 and Eaux under ctrl and cold conditions, (**b**) in Col-0 and Herr under ctrl and hot conditions, and (**c**) in Col-0 and Grip under ctrl and saline conditions. (**) *p*-value < 0.001 of the Student’s *t*-test of the pairwise comparisons between two treatments in each population (black) or between Col-0 and the Pyrenean population at each treatment (colored). The error bars correspond to the standard error of means (SEM).

**Figure 7 ijms-23-03960-f007:**
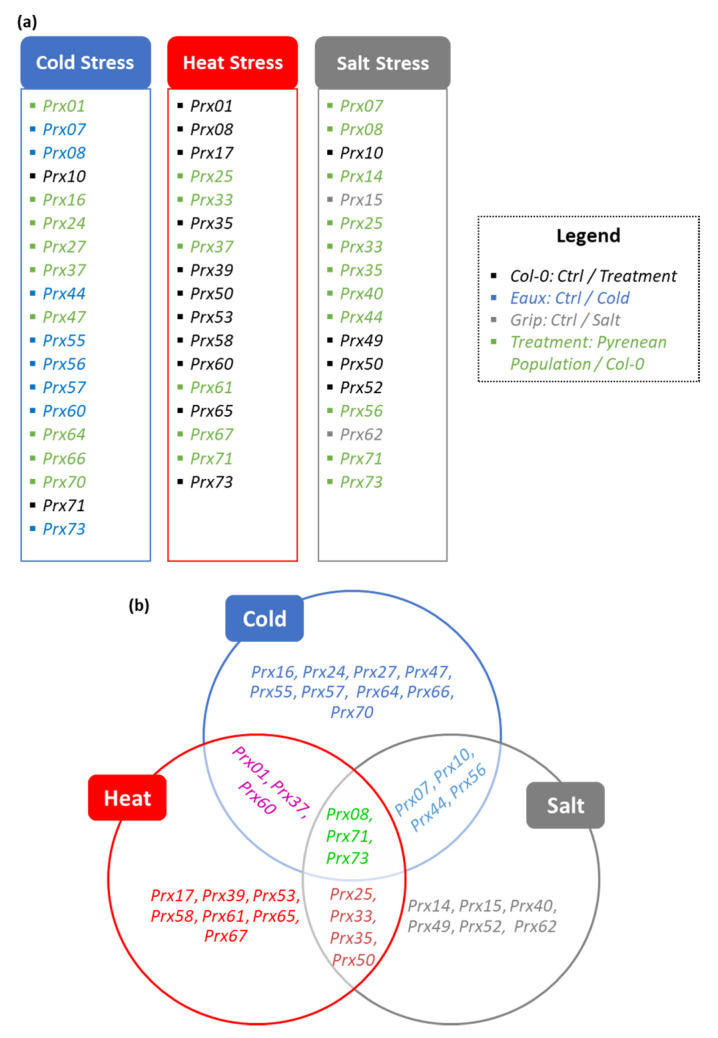
(**a**) Lists of significantly upregulated *CIII Prxs* under cold, heat, and salt stresses that were highlighted by the DE analyses, including genes that were identified by treatment-based comparisons to the ctrl condition in Col-0 (black), by treatment-based comparisons of cold stress to the control in Eaux (blue) and of salt stress to the control in Grip (grey), and by genotype-based comparisons of a Pyrenean population to Col-0 under the different treatments (green). (**b**) Functional specification of different CIII Prxs in response to various abiotic stresses that were identified in Col-0 (not-highlighted) and the Pyrenean populations (green highlight).

## Data Availability

The data presented in this study are available on request from the corresponding author.

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
