# Peer review of "Class III Peroxidases in Response to Multiple Abiotic Stresses in Arabidopsis thaliana Pyrenean Populations"

_ijms, 2022, doi:10.3390/ijms23073960_

Round 1
Reviewer 1 Report
In this manuscript entitled "Class III peroxidases in response to multiple abiotic stresses in Arabidopsis thaliana Pyrenean populations", Eljebbawi et al. suggested that Class III peroxidases play the novel and specific roles in plant tolerance against abiotic stresses. The experimental results are complete and valuable.
The main comments:
Class III PRX is a plant specific subtype, However, I still suggest that author describes the PRX subfamily composition in Introduction. Why the authors studied Class III and not others.
Line42-43, please indicating the specific family members which are involved in abiotic stresses.
Line325, how the authors distinguish PRX activity between different subfamilies.
In the Discussion section, it is necessary to discuss the commonalities and differences between subfamily 3 and other subfamilies proposed in this study in regulating plant stress resistance and growth.
Author Response
We would like to thank the reviewer for the efficient reviewing process and the constructive comments.
Reviewer 2 Report
The submitted manuscript reports on response to multiple abiotic stresses in of Arabidopsis thaliana Pyrenean populations and the role of Class III peroxidases in reactive oxygen species (ROS) regulatory network. The role of several CIII Prxs it is significant in providing tolerance to plants against abiotic stresses as salt tolerance, cold acclimation, and heat tolerance, all in the context of climate changes.
The manuscript it is well-structured, the experiment it is complex and well-designed, focusing on different aspects starting with the root’s phenotype, germination test, transcriptomic and genomic, all supported by a complex statistical analysis. The results are clearly and in a comprehensive manner presented, as well as the methods. All the approached methods have a clear contribution to demonstrate the experimental hypotheses.
Author Response
We would like to thank the reviewer for the efficient reviewing process and the very positive comments.
